# Detection of Interleukin-1 β (IL-1β) in Single Human Blastocyst-Conditioned Medium Using Ultrasensitive Bead-Based Digital Microfluidic Chip and Its Relationship with Embryonic Implantation Potential

**DOI:** 10.3390/ijms25074006

**Published:** 2024-04-03

**Authors:** Tian-Chi Tsai, Yi-Wen Wang, Meng-Shiue Lee, Wan-Ning Wu, Wensyang Hsu, Da-Jeng Yao, Hong-Yuan Huang

**Affiliations:** 1Department of Obstetrics and Gynecology, Chang Gung Memorial Hospital, Linkou Medical Center, 5 Fu-Shin Street, Taoyuan 33301, Taiwan; mp2186@cgmh.org.tw (T.-C.T.); iwen0711@cgmh.org.tw (Y.-W.W.); wlw1830@gmail.com (W.-N.W.); 2Institute of Molecular Medicine and Bioengineering, National Yang Ming Chiao Tung University, Hsinchu 30010, Taiwan; mslee@nycu.edu.tw; 3Department of Mechanical Engineering, National Yang Ming Chiao Tung University, Hsinchu 30010, Taiwan; whsu@nycu.edu.tw; 4Mechanical and Mechatronics System Research Laboratories, Industrial Technology Research Institute, Hsinchu 310401, Taiwan; djyao@itri.org.tw; 5Department of Power Mechanical Engineering, National Tsing Hua University, Hsinchu 30013, Taiwan; 6Department of Obstetrics and Gynecology, College of Medicine, Chang Gung University, Taoyuan 33305, Taiwan

**Keywords:** in vitro fertilization, implantation, embryo selection, cytokine

## Abstract

The implantation of human embryos is a complex process involving various cytokines and receptors expressed by both endometrium and embryos. However, the role of cytokines produced by a single embryo in successful implantation is largely unknown. This study aimed to investigate the role of IL-1β expressed in a single-embryo-conditioned medium (ECM) in embryo implantation. Seventy samples of single ECM were analyzed by a specially designed magnetic-beads-based microfluidic chip from 15 women. We discovered that IL-1β level increased as the embryo developed, and the difference was significant. In addition, receiver operator characteristic (ROC) curves analysis showed a higher chance of pregnancy when the IL-1β level on day 5 ECM was below 79.37 pg/mL and the difference between day 5 and day 3 was below 24.90 pg/mL. Our study discovered a possible association between embryonic proteomic expression and successful implantation, which might facilitate single-embryo transfer in the future by helping clinicians identify the embryo with the greatest implantation potential.

## 1. Introduction

Implantation is a multifaceted process characterized by intricate interactions between the endometrium and embryos [1,2]. The endometrium undergoes significant morphological and functional changes to support the implantation of a competent embryo, orchestrated by a multitude of growth factors, cytokines, chemokines, adhesion molecules, and sex steroids secreted by both the endometrium and blastocysts [3,4]. However, preclinical pregnancy wastage in humans can be as high as 30%, highlighting the importance of a coordinated maternal–fetal dialogue [5]. Numerous studies have sought to understand the detailed mechanisms to improve artificial reproductive outcomes [2]. Researches indicated that developmentally competent embryos created a distinct microenvironment and triggered specific responses in the endometrium compared to developmentally compromised embryos [6,7]. Developmentally competent human embryos induced Ca^2+^ signaling in the luminal cells and initiate the gene expression related to implantation, whereas developmentally compromised embryos induced endoplasmic reticulum stress, leading to implantation failure. This study revealed that the embryos forecasted their developmental competence before implantation, and the microenvironment created by the embryo itself implicated its competence [7].

Selecting the embryo with the greatest implantation potential for transfer is a critical step toward successful pregnancy in artificial reproductive technology cycles [8,9]. Morphologic assessment, such as a blastocyst scoring system developed by Gardner et al., is commonly used to select the optimal embryos [10]. However, with recent advancements in translation technology, several biomarkers have been shown to reflect embryo viability [11,12]. Cytokines such as interleukin 1β (IL-1β), interleukin 6, and tumor necrosis factor-α have also been shown to be involved in embryo–endometrium dialogue in the implantation process [1,13]. These proteomic profiles of embryos provide crucial information regarding their developmental and implantation potential.

We have previously demonstrated the presence of the IL-1β system in the microenvironment created by single preimplantation embryos. We have demonstrated that embryonic IL-1β expression increases progressively with advancing embryonic development, reaching its peak levels in hatching blastocysts [14]. It regulates trophoblast invasion and participates in immunologic reaction such as natural killer cell and T lymphocyte activation [15]. However, IL-1β may have beneficial as well as adverse effects on reproductive cells, likely depending on its local concentration, and the IL-1β overexpression is associated with recurrent pregnancy loss [16,17]. These studies suggested that IL-1β was a key factor in embryonic implantation, but excessive expression might lead to adverse pregnancy outcomes.

The embryo culture medium contains valuable information about the competence of embryos, yet analyzing these media poses technical challenges. However, with advancements in molecular science, we now have the capability to delve deeper into the intricate microenvironment established by embryos. The examination of medium from individual embryos is especially critical, offering vast benefits for the precision medicine field in clinical settings. Compared to conventional enzyme-linked immunosorbent assay (ELISA), digital microfluidic immunoassay provides several advantages, including small sample volume requirement, rapid reaction time, and low detection limits [18]. A magnetic-beads-based digital microfluidic immunoassay developed by our group can further achieve the lower detection limit and improve sensitivity by aggregating beads at the detection stage. With the use of this magnetic-beads-based digital microfluidic immunoassay chip, which requires only 520 nL of sample and achieved a detection limit of 0.0287 pg/mL [18], the cytokines level from a single-embryo-conditioned medium can be measured.

In the present study, we used a magnetic-beads-based digital microfluidic immunoassay chip to investigate the relationship between pregnancy and the level of IL-1β in a single-embryo-conditioned medium.

## 2. Results

### 2.1. Patient Characteristics and Reproductive Outcomes

A total of 15 patients were included. The mean age of the study population was 33 ± 6 years, the mean BMI was 21.78 ± 2.33 kg/m^2^, and the mean level of anti-Müllerian hormone was 5.90 ± 4.15 ng/mL. The etiologies of infertility includes ovulatory dysfunction (n = 12), tubal factor (n = 3), and male factor (n = 7). Progestin-primed ovarian stimulation protocol or gonadotropin-releasing hormone (GnRH) antagonist protocol was used depending on the clinician’s decision. The mean number of retrieved oocytes was 20 ± 6, with average blastocysts being 9 ± 5. Nineteen embryo transfer cycles were performed, resulting in 8 pregnancies with 1 chemical pregnancy and 7 clinical pregnancies. The patient demographics and reproductive outcomes are shown in Table 1.

### 2.2. Analyzing Embryo-Conditioned Sample by a Magnetic-Beads-Based Digital Microfluidic Immunoassay Chip

The magnetic-beads-based digital microfluidic immunoassay chip conducts immunoassay steps through the manipulation of droplets and magnetic beads [18]. The droplet manipulation is achieved through electrowetting-on-dielectric (EWOD) technology, with the chip featuring an electrode array for the generation, transfer, and mixing of solution droplets. The design of the electrodes enables the requirement for only a small sample volume, needing just 520 nL of the sample [18].

Figure 1 illustrates the layout of the immunoassay chip. On the left side, there are five reservoirs for loading the beads solution, sample solution, detection antibody solution, fluorescent reporter solution, and washing buffer, respectively, as shown in Figure 1a. On the right side, there are five larger major electrodes, used to generate droplets from the corresponding reservoirs. Between those major electrodes, there are smaller electrodes used to generate smaller daughter droplets containing magnetic beads from the major electrode, which are then transferred to the next major electrode for immunoreaction steps.

The immunoreaction process comprises five sequential steps. Initially, target capture occurs by mixing a droplet with magnetic beads with the sample for 12 min (Figure 1(b1)). Next, detection antibodies are captured by blending a similar droplet with an antibody droplet for 6 min (Figure 1(b2)). The third step involves mixing with a fluorescent reporter droplet for 6 min (Figure 1(b3)). This is followed by a washing step (Figure 1(b4)) and finally, fluorescence measurement, where the bead-containing droplet is moved to a specific detection area (Figure 1(b5)). In each immunoreaction, the same quantity of magnetic beads (100 beads) is consistently used to maintain a low coefficient of variation (CV) [18]. Throughout these processes, a fixed number of magnetic beads is maintained to prevent loss. Additionally, before measuring fluorescence, the beads are aggregated to enhance the intensity of the fluorescence.

### 2.3. Beads, Reagents, and Sample Preparation

The utilized magnetic beads were COOH-modified and had a diameter of 6 μm (COMPEL™, Bangs Laboratories, Inc., Fishers, IN, USA). The captured antibodies, referring to the human IL-1β antibody, were sourced from R&D Systems, Minneapolis, MI, USA. The process of conjugating the captured antibody to the magnetic bead was performed by MagQu Co., Ltd., New Taipei, Taiwan, according to a modified protocol. For establishing the calibration curves, recombinant human IL-1β antigens from R&D Systems, USA, were used. The fluorescent reporter was R-PE Streptavidin, from R&D Systems, USA. The absorption and emission wavelengths of the fluorescence fell in the range of 488–494 nm and 515–545 nm, respectively. Both Pluronic F-127 and BSA were also sourced from R&D Systems, located in the USA.

A detection antibody solution for human IL-1β, at a concentration of 20 μg/mL, was prepared using PBS mixed with PL (Pluronic F-127 in PBS at 6 mg/mL). The fluorescent reporter was initially prepared as a 2 mg/mL stock solution with DI water, which was then diluted 100-fold using PL and PBS (Pluronic F-127 in PBS at 6 mg/mL) to create the final fluorescent reporter solution.

### 2.4. Establishing the Calibration Curve for Human Interleukin 1-β and Data Collection

In establishing the calibration curve for human IL-1β, various known concentrations of human IL-1β samples were selected, spanning from 0 pg/mL to 100 pg/mL, including intermediate concentrations such as 0.1 pg/mL, 1 pg/mL, 10 pg/mL, 25 pg/mL, 50 pg/mL, and 75 pg/mL. The standard curve is illustrated in Figure 2, with 100 beads used. In our previous studies, it has been shown that the coefficient of variation (CV) was lowest when 100 beads were utilized, compared to instances where 25 and 49 beads were employed [18]. This finding underscores the superior reproducibility associated with using 100 beads. The calibration curve underwent three repetitions of experiments at every concentration level. In addition, we further adjust the microscope setting parameters, as shown in Figure 3. The purpose of adjusting parameter settings is to optimize the measurement capability. As shown in Figure 3, different parameter settings have different detection limits and show different fluorescence intensity at a given concentration. When the fluorescence microscope setting is 100, it has lower detection limits compared to parameters set as 80 (blue) and 90 (orange). In addition, when the parameter settings are 110 (yellow), 120 (blue), and 130 (green), the differences in intensity between different concentrations were quite small, so chances of measurement error are bigger compared to 100 (red). Consequently, in subsequent analyses of human embryo condition medium samples, 100 beads and the fluorescence microscope parameter setting as 100 were selected for use, given their demonstrated reliability and precision in quantitative analysis.

### 2.5. Interleukin 1-β and Embryonic Development and Implantation Potential

The mean values of IL-1β from the ECM on days 3 and 5 were evaluated and are presented in Figure 4 and Table 2. However, due to the extremely low concentration of IL-1β, some of the measured levels were exceptionally high or low. As a result, 18 samples were excluded from our data base. The reasons for exclusion are as follows: 3 pairs of day 3 and day 5 were excluded because the IL-1β concentration on day 5 was much lower compared to other pairs; another 3 pairs were excluded due to the difference in IL-1β concentration between days 5 and 3 being markedly different from others; 1 pair was excluded because the IL-1β concentration on day 5 was significantly higher; and 2 pairs were excluded due to a much lower IL-1β concentration on day 3. After the extreme values are deleted, in a total of 52 samples, the mean day 3 IL-1β level is 47.18 ± 19.45 pg/mL, and the day 5 IL-1β level is 79.64 ± 30.75 pg/mL. There is a significant increase in the level of IL-1β in day 5 ECM compared to day 3 ECM by paired *t*-test (*p* < 0.05). The trend is also observed both in the non-pregnant group and pregnant group (*p* < 0.05), as shown in Figure 5. There is no significant difference in the mean value of IL-1β from the day 3, day 5, and day 5–day 3 ECM difference in non-implanted groups and implanted groups, as shown in Table 2. However, a slightly lower degree of increase in day 5 IL-1β is observed in the pregnant group, with the difference being 21.54 ± 22.80 vs. 37.86 ± 27.18 in the non-pregnant group.

### 2.6. Interleukin 1β Levels as a Diagnostic Tool to Predict Successful Implantation

We further analyzed the diagnostic performance of IL-1β levels as a tool to predict pregnancy outcomes. We conducted a thorough analysis to assess the diagnostic efficacy of IL-1β levels in predicting pregnancy outcomes. The results, depicted in Figure 6, highlight the values of the area under the curve (AUC) in the receiver operating characteristic (ROC) curves. Specifically, IL-1β levels in day 5 ECM exhibited an AUC of 0.688. Upon further analysis, we identified an optimal cutoff point of 79.37 pg/mL, with a sensitivity of 78.6% and a specificity of 72.7%. These findings suggest that IL-1β levels below 79.37 pg/mL are associated with a higher likelihood of successful pregnancy. Additionally, we evaluated the difference in IL-1β levels between day 5 ECM and day 3 ECM. This comparison yielded an AUC of 0.701. The optimal cutoff point was determined to be 24.90 pg/mL, with a sensitivity of 85.7% and a specificity of 63.6%. Consequently, a difference in IL-1β levels below 24.90 pg/mL indicates a better chance of achieving pregnancy.

## 3. Discussion

Various methods exist to assist clinicians in selecting the most optimal embryos for transfer. Traditionally, embryologists have relied on grading embryos based on their morphology, often employing the criteria established by Gardner et al. [10]. This involves assessing factors such as blastocyst expansion, size, inner cell mass, and trophectoderm. However, with the introduction of time-lapse systems in artificial reproduction technology, studies have found that several factors beyond morphology, such as fertilization markers, cleavage stage features, and morphokinetic parameters, significantly impact implantation rates and live birth rates [19,20]. Consequently, the adoption of time-lapse systems has emerged as a valuable tool for identifying embryos with favorable prognoses. Additionally, preimplantation genetic screening (PGS) using next-generation sequencing enables the determination of an embryo’s ploidy status, offering clinicians further insights into its viability. A recent study incorporating results from IDAscore, an AI-based scoring system derived from time-lapse data, PGS data, and maternal uterine pulsatility index, demonstrated enhanced predictive capabilities for live birth rates when these three parameters were combined. Specifically, the combined approach achieved an area under curve of 0.67, surpassing the predictive accuracy of standalone time-lapse scoring systems or traditional morphology assessments [21]. While these findings are promising, a significant gap persists between identifying the most viable embryo and the current assessment methods.

The connection between the proteomic expression of human embryos and pregnancy outcomes has been explored in several studies [22,23,24]. Dominguez et al. conducted a study where they analyzed the pooled blastocysts-conditioned medium using Chemiarray™. This analysis revealed different cytokine expression between implanted and non-implanted embryos. Out of the 120 proteins analyzed, including TNF-α, IL-1β, GM-CSF, CXCL13, and IL-6, only a decrease in GM-CSF and CXCL13 were observed in non-implanted embryos [22]. Biba et al. utilized commercially available ELISA kits (Human IL-1β and Human IL-6 Singleplex Bead Kit; Invitrogen Carlsbad, CA, USA) to analyze IL-1β and IL-6 at the blastocyst stage in a single ECM. They found that 26.5% of the samples had detectable levels of IL-1β, but no correlation was observed between pregnancy outcomes and the detectability of IL-1β [24]. Dominguez et al. conducted another study incorporating time-lapse and proteome analysis of embryos and suggested that the presence of IL-6 and 5–12 h cc2 had significantly higher implantation rates [11]. These studies utilized commercial kits but failed to show a conclusive relationship between pregnancy and these cytokines. Moreover, these studies did not provide detailed information on the levels of these factors, and it is worth noting that the detection limits of commercially available kits for these cytokines may not accurately reflect the intricate relationship between the cytokines and implantation process.

IL-1β is a pro-inflammatory cytokine, and its level has been shown to increase as embryos develop and reach highest expression in the blastocyst stage in previous studies [14,25,26,27,28]. Furthermore, it has also been shown that exogenous supplement of IL-1β in mouse embryo culture medium was associated with better blastocysts quality, blastocysts hatching percentage, and faster hatching speed [29,30]. In addition to embryo development, IL-1β also plays a key role in regulating the inflammatory process at the embryo–maternal interface [31,32]. It causes endometrial transformation, stimulates chemokines secretion from natural killer cells, regulates trophoblast invasion, activates matrix-degrading metalloproteinases, and promotes blastocyst motility in vitro [15,32,33]. Furthermore, research has discovered that IL-1β also participated in immunological regulation of the T helper 1 to T helper 2 ratio. An imbalance in this ratio has been linked to various adverse pregnancy outcomes, including pregnancy loss, preeclampsia, and preterm labor [34,35,36]. However, higher IL-1β is found to be associated with recurrent pregnancy loss, suggesting that IL-1β may have both beneficial and harmful effects on implantation [17,37]. These studies suggest that IL-1β is important in embryo development and early implantation, but a balanced expression is needed for successful pregnancy.

The advancement in biomedical technology has enabled scientists to investigate the proteomic expression of embryos [38]. Conventional ELISA requires at least 20 μL of sample volume, so measuring the analyzed concentration in embryo-conditioned medium is challenging. Previous studies use pooled conditioned medium or reported the presence of certain cytokines. With the use of a magnetic-bead-based digital microfluidic chip, the concentration of a substrate in a single-embryo culture medium can be analyzed. Furthermore, the detection sensitivity can be magnified by manipulating the magnetic beads aggregation [38].

In our study, we delved into the proteomic expression of interleukin-1 beta (IL-1β) within a single-embryo-conditioned medium. Building upon prior research demonstrating an increase in IL-1β expression at the mRNA level during embryo development, we demonstrated a corresponding elevation in IL-1β expression at the proteomic level [14,27]. Notably, there exists an optimal concentration range of IL-1β during the blastocyst stage, crucial for successful implantation, highlighting the potential detrimental effects of IL-1β overexpression and the necessity for balanced secretion. The unique contribution of our study lies in being the first to elucidate the correlation between IL-1β levels in single-embryo-conditioned medium and implantation success. With the help of a magnetic-bead-based digital microfluidic chip, it allowed us to gain a deeper understanding of the relationship between IL-1β and implantation, and this valuable information may be added to current methods of selecting embryos.

Our study is subject to several limitations. Firstly, the sample size is relatively small, consisting of only 15 patients with a total of 70 samples included, and 18 samples were not included in the final statistical analysis due to extreme values. Secondly, due to the limited availability of human embryos, the selection of embryos for transfer was based on morphology assessment rather than IL-1β levels. Consequently, the analysis of IL-1β levels was conducted retrospectively after embryo transfer, potentially introducing biases in the interpretation of results.

## 4. Materials and Methods

### 4.1. Study Population and Clinical Management

A total of 15 patients were included between 2018 and 2020 in the study (aged 23–41 years and BMI between 17.9 and 27.2 kg/m^2^). Controlled ovarian hyperstimulation was conducted using GnRH antagonist or progestin-primed ovarian protocol. On day 2 or day 3 of the menstruation cycle, laboratory data including follicle-stimulating hormone, luteinizing hormone, and estradiol were measured, and transvaginal ultrasound was performed to evaluate follicular number and size. Gonadotropins were initiated also on day 2 or day 3 with a dosage of 150–300 IU daily, and the medication included recombinant-follicle-stimulating hormone (r-FSH) combined with recombinant-luteinizing hormone (r-LH, Pergoveris^®^, 150 IU/75 IU/vial, Merck Serono, Fenil-sur-Corsier, Switzerland), recombinant-follicle-stimulating hormone (r-FSH, Gonal-F^®^, 5.5 mcg/vial, Merck Serono, Switzerland), human menopausal gonadotrophin (HMG, Menopur^®^, 75 IU/75 IU/vial, Ferring, Kiel, Germany), or long-acting r-FSH (Elonva^®^ 100 μg/0.5 mL or 150 µg/0.5 mL, Vetter Pharma-Fertigung, Ravensburg, Germany). Follicular size and hormone levels were later measured on day 6, and the gonadotropins dosage would be adjusted accordingly. GnRH antagonist (Cetrotide^®^, 0.25 mg/vial, Merck Serono, Switzerland) or dydrogesterone film-coated tablet (Duphaston, 10 mg, Abbott, Hoofddorp, The Netherlands) was used for the inhibition of premature LH surge. When the dominant follicle reached 16–18 mm, dual triggering with 0.2 mg of triptorelin acetate (Decapeptyl, 0.1 mg/mL, Ferring, Germany) and 250 mcg of choriogonadotropin alfa (Ovidrel, 250 mcg, Merck Serono Italy, Roma, Italy) was given. Oocyte aspiration was done 34–36 h later and subject to in vitro fertilization or intracytoplasmic sperm injection. On day 3, the cleavage stage embryos were first evaluated by embryologists and the assessments include cell number, cell symmetry, and cellular fragmentation using Lucinda Veek’s grading system [39]. Figure 7 is an example of a cleavage stage embryo. The embryos were later assessed on day 5 by an embryo grading system established by Garder et al. [10] for transferring, which includes the evaluation of expansion status rating on a scale of 1 to 6, trophectoderm quality grading as A, B, or C according to its thickness and arrangement, and inner cell mass grading as A, B, or C based on factors like cell number and compactness. Figure 8 is an example of transferred embryos. The conditioned medium of a transferred embryo was retrospectively analyzed by a magnetic-beads-based digital microfluidic immunoassay chip. Seventy samples with 35 samples from culture day 3 and 35 samples from culture day 5 were analyzed. Successful implantation was considered when serum β-hCG was above 5 mIU/mL.

### 4.2. Statistics

Paired *t*-test was used to compare day 3 and day 5 IL-1β concentration. The embryos were further divided into two groups according to their implantation status. Nonparametric statistics test was used to compare the IL-1β concentration in the two groups due to the small sample size. Significance was considered when *p* < 0.05. The receiver operating characteristic curve (ROC) was further used to analyze the diagnostic performance of embryonic IL-1β. An area under the ROC curve >0.5 might indicate that IL-1β concentration could be used as a tool to identify embryos with a better chance of implantation.

## 5. Conclusions

In conclusion, our study demonstrates that IL-1β concentration increases as the embryo develops, and the degree of increase is negatively associated with pregnancy. As technology advances, more substrates may be targeted, providing more information about embryo viability. Current technologies such as time-lapse systems and PGS hold promise for the future of embryo selection. Our studies suggest that the proteomic profile of an embryo could also offer valuable information regarding its viability. In the future, combining these proteomic data with traditional morphology assessments, time-lapse systems, and PGS may enhance the likelihood of successful implantation during single-embryo transfer. This could avoid multiple gestation, which carries higher maternal and fetal risks. However, more extensive randomized trials are needed to better understand the role of embryonic proteomic expression in the field of reproductive biology.

## Figures and Tables

**Figure 1 ijms-25-04006-f001:**
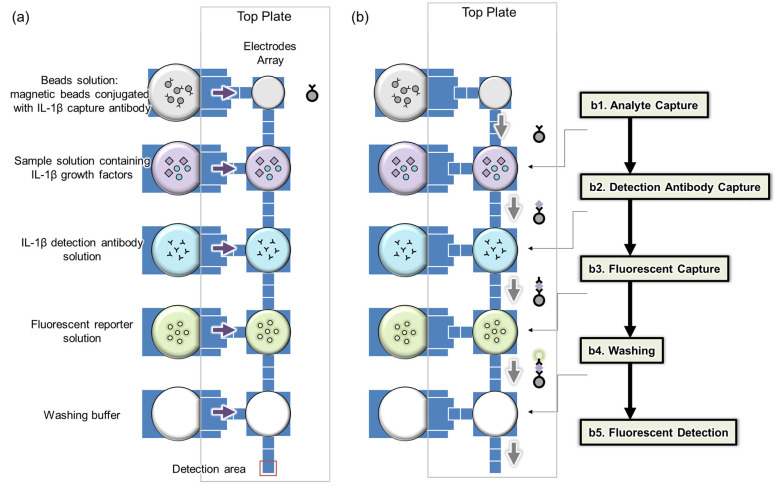
Analysis using a bead-based digital microfluidic immunoassay chip: (**a**) electrode array and reagent loading, (**b**) on-chip analysis procedures.

**Figure 2 ijms-25-04006-f002:**
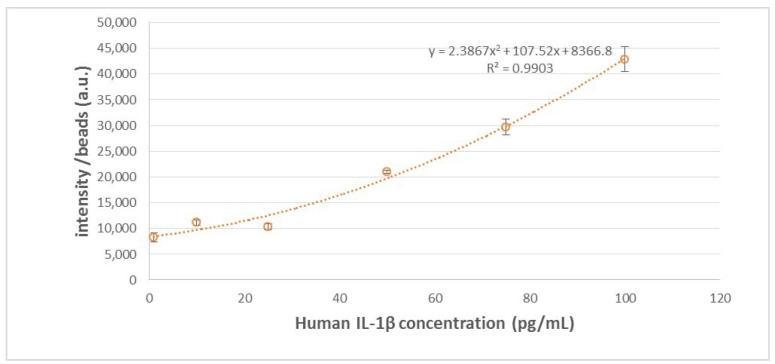
IL-1β standard curve with 100 beads used.

**Figure 3 ijms-25-04006-f003:**
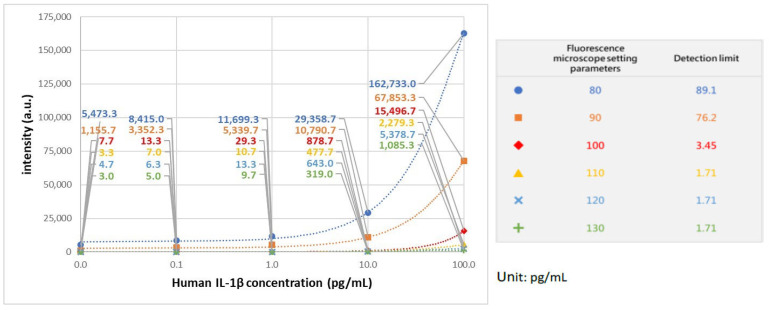
IL-1β standard curve with different fluorescence microscope setting parameters.

**Figure 4 ijms-25-04006-f004:**
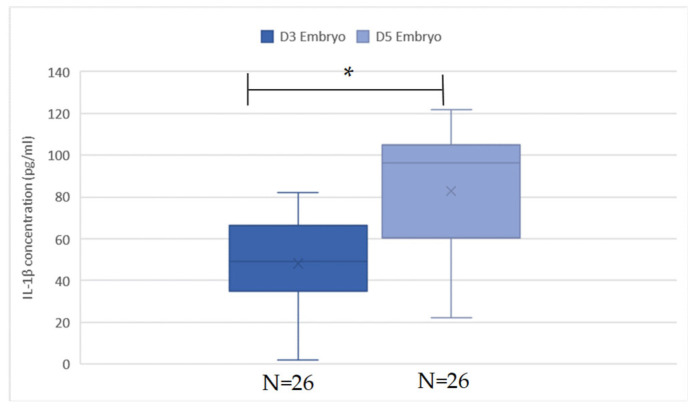
IL-1β value in D3 and D5 embryos. The concentration of IL-1β was significantly higher in the group of D5 embryos when analyzed by paired *t*-test (*p* < 0.05). * is significant.

**Figure 5 ijms-25-04006-f005:**
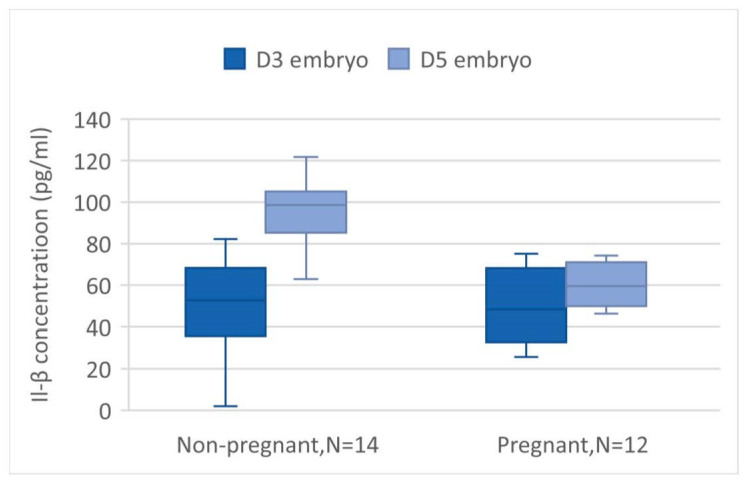
The IL-1β value in D3 and D5 embryos was compared in subgroups according to pregnant or non-pregnant. The difference is significant in both groups (*p* < 0.00 and <0.01 in non-pregnant and pregnancy groups).

**Figure 6 ijms-25-04006-f006:**
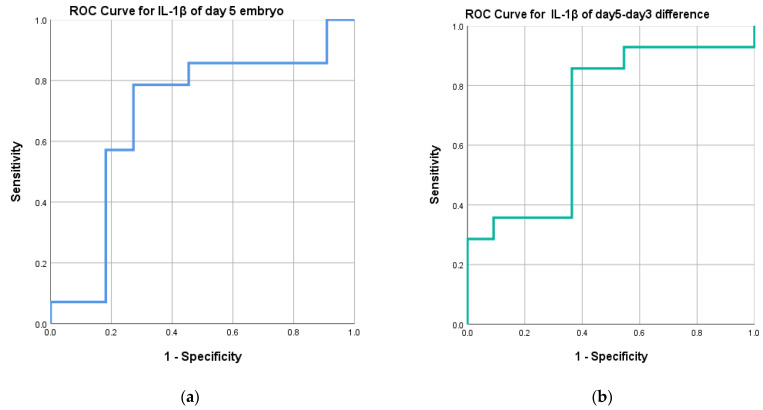
ROC curves: (**a**) RCO curve for day 5 ECM IL-1β levels, (**b**) RCO curve for day 5–day 3 ECM IL-1β levels.

**Figure 7 ijms-25-04006-f007:**
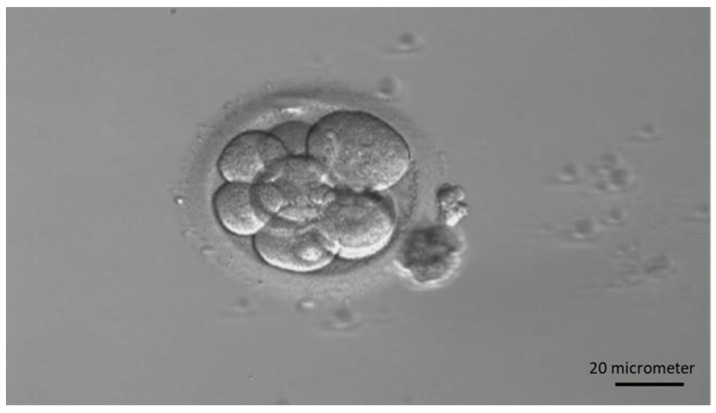
Example of a cleavage stage embryo under 20× magnification.

**Figure 8 ijms-25-04006-f008:**
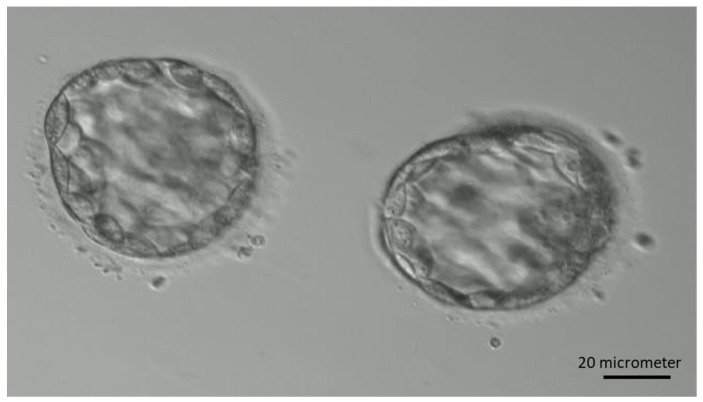
Example of two transferred blastocysts under 20× magnification.

**Table 1 ijms-25-04006-t001:** Patient demographics and reproductive outcomes.

Characteristics	N = 15
Age (years) (Mean ± SD)	33 ± 6
BMI (kg/m^2^) (Mean ± SD)	21.78 ± 2.33
AMH (ng/mL) (Mean ± SD)	5.90 ± 4.15
Day 3 FSH (mIU/mL) (Mean ± SD)	6.00 ± 1.64
GnRH antagonist protocol, n	11
PPOS protocol, n	4
E2 on triggering day (pg/mL) (Mean ± SD)	3190 ± 2335
Retrieved oocytes number (Mean ± SD)	20 ± 6
Number of 2PN zygotes (Mean ± SD)	11 ± 6
Number of blastocysts (Mean ± SD)	9 ± 5

**Table 2 ijms-25-04006-t002:** Mean value of IL-1β with respect to pregnant or not.

	Non-Pregnant (N = 14)	Pregnant (N = 12)
Day 3 IL-1β (pg/mL)	49.46 ± 21.58	46.49 ± 16.52
Day 5 IL-1β (pg/mL)	88.15 ± 28.69	68.04 ± 30.91
Day 5–Day 3 IL-1β (pg/mL)	37.86 ± 27.18	21.54 ± 22.80

## Data Availability

The data presented in this study are available on request from the corresponding author. The data are not publicly available due to containment of patient identifiers.

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
