# Peer review of "Detection of Interleukin-1 β (IL-1β) in Single Human Blastocyst-Conditioned Medium Using Ultrasensitive Bead-Based Digital Microfluidic Chip and Its Relationship with Embryonic Implantation Potential"

_ijms, 2024, doi:10.3390/ijms25074006_

Round 1
Reviewer 1 Report
Comments and Suggestions for Authors
The presented paper concerns the very important problem of assessing the quality of human embryos in ART cycles along with the classical morphological assessment routinely used in ART clinics. The authors developed a new digital microfluidic immunoassay based on magnetic beads that allows the assessment of IL-1β level in a single-embryo conditioned medium. It is important to note that this method, as detailed in the Discussion section, allows for the use of a very small sample and is characterized by fast reaction times and low detection limits for IL-1β. In the context of comparing the implantation potential, the authors concluded that IL-1β concentrations increase as the embryo develops, with the degree of increase negatively correlated with pregnancy. Some limitations of the study that the authors point out are appreciated. The article is concise and clearly written. Overall, I am impressed with the work of this group of scholars and can recommend the article for publication after minor revisions.
Thereafter, I have only a few minor comments for the authors.
1. In lines 247–248, the authors state that the embryos were assessed by morphological criteria for transferring. I think it would be helpful to list these criteria somewhere in the paper—perhaps in an appendix/supplement. It would be great to provide photomicrographs of blastocysts that can/cannot be transferred, although I do not insist on this. Perhaps these criteria are obvious for clinical embryologists, but for a wide range of readers, it seems to me that they would be useful in this article.
2. In the Results section, the references to the figures, as well as the numbers of the figures, are confused (for example, different figures 4 appear twice). Please check all figure references carefully and make appropriate corrections to the text, starting at line 125.
3. Please remove any extra italics throughout Section 4.4.
Author Response
Dear reviewer :
Thank you very much for taking the time to review this manuscript. Please find the detailed responses below
Comment 1. In lines 247–248, the authors state thatthe embryos were assessed by morphological criteria for transferring.I think it would be helpful to list these criteria somewhere in the paper—perhaps in an appendix/supplement. It would be great to provide photomicrographs of blastocysts that can/cannot be transferred, although I do not insist on this. Perhaps these criteria are obvious for clinical embryologists, but for a wide range of readers, it seems to me that they would be useful in this article.
Response:
Thank you for the comments. I have added some description of the blastocyst morphology assessment developed by Gardner et al and Lucinda's grading system for cleavage stage embryos. I also added two pictures of different stage embryos.
Comment 2 : In the Results section, the references to the figures, as well as the numbers of the figures, are confused (for example, different figures 4 appear twice).Please check all figure references carefully and make appropriate corrections to the text, starting at line 125.
Response:
Yes, I have made the correction
Comment 3. Please remove any extra italics throughout Section 4.4.
Response: Yes, I have made the correction
Thank you for your comments
Reviewer 2 Report
Comments and Suggestions for Authors
The study utilized an ultrasensitive bead-based digital microfluidic chip to measure interleukin-1 β (IL-1β) levels in the conditioned medium of single human embryos, finding that IL-1β levels increase with embryonic development. The authors discovered a potential link between lower IL-1β levels in embryo-conditioned medium on day 5 and increased pregnancy chances, suggesting IL-1β as a biomarker for embryo selection in in-vitro fertilization. This research could aid in optimizing single embryo transfer to improve pregnancy success rates in assisted reproductive technologies.
The paper is of good quality, without clear flaws or weaknesses. Here are some specific comments.
- Figure 4, the microfluidic chip and illustration, should move to figure 1. It is canonical to first introduce the system.
- Then it is nicely followed by Figure 1, which is the calibration/validation data. Regarding Figure 1, the first two data points have no error bars, which could be because of formatting issues and should be an easy fix.
- More interpretation needs to be made about Figure 2, is the greenline (130) even functional? Why did the authors do this experiment and what is the conclusion?
- More discussion is needed comparing this to current gold standard methods, is this doing better?
